# 3D Priors-Guided Diffusion for Blind Face Restoration

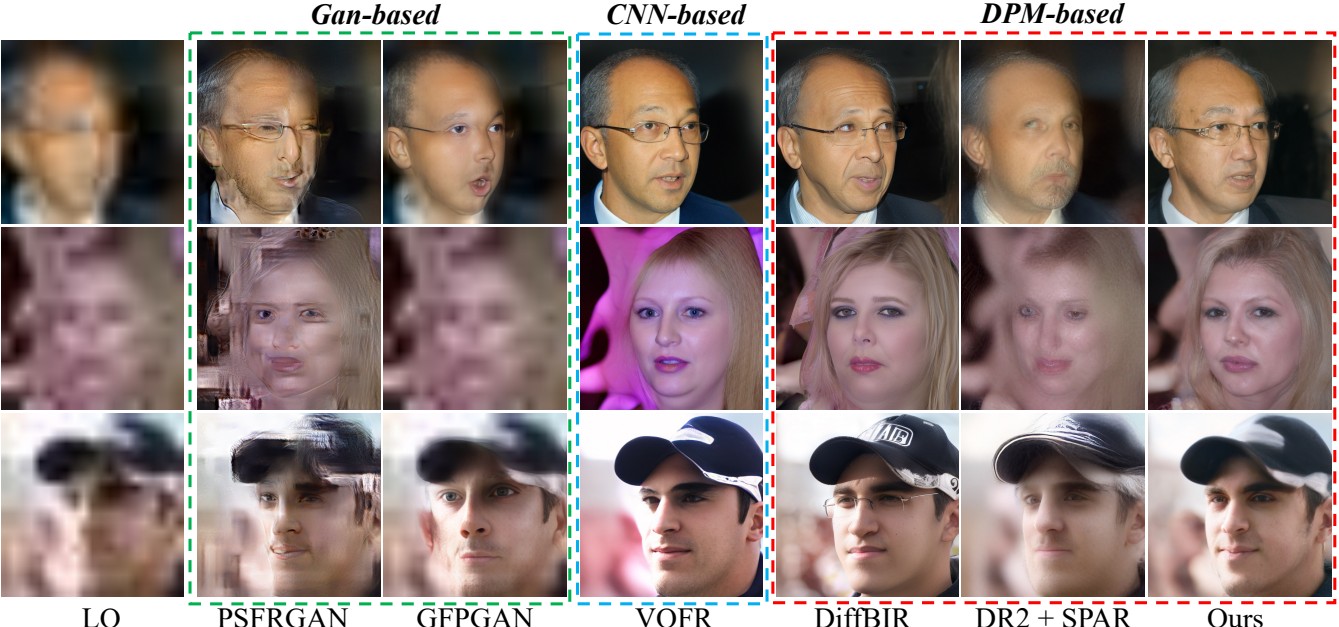

**Figure 1: Comparison of our method with other blind face restoration methods. The left segment illustrates low-quality images (LQ), while the center showcases the outcomes of Generative Adversarial Networks (GANs) in the green rectangle [3, 37], results from a CNN-based model in the blue region [11], and outcomes from the Diffusion Probability Model (DPM) in the red rectangle [25, 40]. Our approach excels in enhancing restoration details significantly.**

## ABSTRACT

Blind face restoration endeavors to restore a clear face image from a degraded counterpart. Recent approaches employing Generative Adversarial Networks (GANs) as priors have demonstrated remarkable success in this field. However, these methods encounter challenges in achieving a balance between realism and fidelity, particularly in complex degradation scenarios. To inherit the exceptional realism generative ability of the diffusion model and also constrained by the identity-aware fidelity, we propose a novel diffusion-based framework by embedding the 3D facial priors as structure and identity constraints into a denoising diffusion process. Specifically, in order to obtain more accurate 3D prior representations, the 3D facial image is reconstructed by a 3D Morphable Model (3DMM) using an initial restored face image that has been processed by a pretrained restoration network. A customized multi-level feature extraction method is employed to exploit both structural and identity information of 3D facial images, which are then mapped into the noise estimation process. In order to enhance the fusion of identity information into the noise estimation, we propose a Time-Aware Fusion Block (TAFB). This module offers a more efficient and adaptive fusion of weights for denoising, considering the dynamic nature of the denoising process in the diffusion model, which involves initial structure refinement followed by texture detail enhancement. Extensive experiments demonstrate that our network performs favorably against state-of-the-art algorithms on synthetic and real-world datasets for blind face restoration.

## CCS CONCEPTS

• **Computing methodologies → Computer vision problems**.

## KEYWORDS

Blind face restoration, Diffusion probabilistic models

## 1 INTRODUCTION

Blind face restoration is a long-standing vision task that involves recovering a high-quality face image from a low-quality observation. It plays an important role in old photo recovery and face recognition.

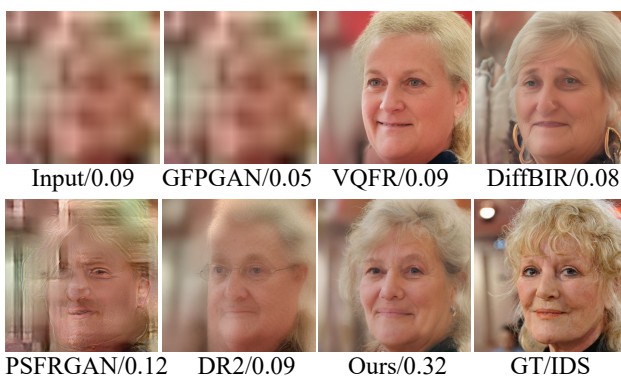

Input/0.09  GFPGAN/0.05  VQFR/0.09  DiffBIR/0.08

PSFRGAN/0.12  DR2/0.09  Ours/0.32  GT/IDS

**Figure 2: Comparison of the performance of different methods on the Arcface identity score (IDS). The restoration results in the first-row show that our method is better consistent with GroundTruth(GT) in the restoration of facial features.**

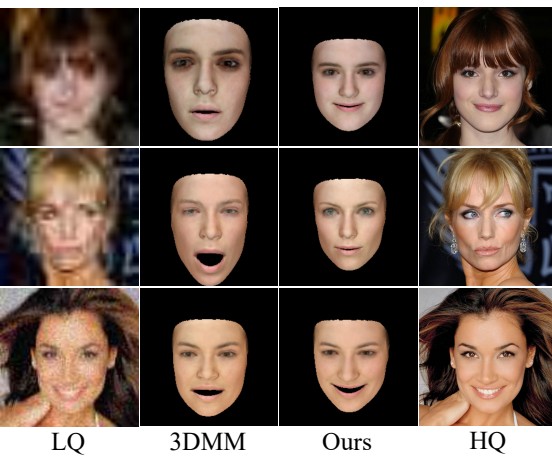

LQ    3DMM    Ours    HQ

**Figure 3: Comparison of reconstructing 3D faces from the 3D Morphable Model (3DMM) and ours. By using SwinIR to perform initial restoration and then reconstruction, we can obtain a better reconstructed 3D face in terms of facial expressions, structure, facial features, mouth shape, and illumination.**

As an ill-posed problem, this task is highly challenging since low-quality face images can suffer from multiple degradations, such as downsampling, blurring, noise, and compression.

Multifarious traditional end-to-end methods have been proposed based on convolutional neural networks (CNNs) to learn mapping relationships between low-quality and high-quality image pairs, but they fail to restore fine details on the face [5, 26]. Recently, due to the powerful ability of Generative Adversarial Networks (GANs) [8] to generate realistic face images, some methods [9, 42, 47] have utilized GANs as a prior for generating high-quality faces. These methods extract low-quality face features using CNNs and encode them into the latent space of the GAN. However, methods employing GANs as priors may encounter training collapse [45], a consequence of challenges in optimizing the objective function during training. To improve the identity consistency of the restored images, some methods incorporate prior information during the face restoration process, such as facial geometric priors [4, 15] and reference priors [22, 23]. For example, Gu *et al.* use vector quantization face Restoration (VQFR) [11] to guide the model for generating more realistic facial details by storing the features extracted from high-resolution images in a dictionary. Some methods [39, 46] leverage the powerful feature extraction capability of Vision Transformer (ViT) [36] to achieve better restoration results. However, as shown in Fig. 2, the restoration results of these models cannot achieve a balance between fidelity and authenticity when dealing with complex scenes or under specific conditions. These approaches continue to face challenges in accurately restoring intricate facial features while simultaneously capturing realistic textural qualities, as exemplified in Fig. 1.

More recently, denoising diffusion models have been introduced into image restoration. For example, [32] feeds the combination of low-quality and noisy images into the denoising diffusion model for noise prediction. [30] encodes the low-quality image once and passes it through a cross-attention mechanism, feeding the features into a diffusion model. Additionally, [25] utilizes a parallel encoding module to encode the condition information and inputs it into the decoder of the denoising diffusion model at each step of the denoising process. SR3 [32] presents a method for achieving image super-resolution through iterative refinement, involving the generation of conditional images by concat low-resolution images and noise images. Latent diffusion [30] learns the data distribution in the latent space by gradually diffusing the noise signal and completes the guidance by applying cross attention to the conditions in the denoising network. However, throughout the denoising diffusion process, the low-frequency components exhibit gradual changes as the time step progresses, whereas the high-frequency components undergo more pronounced alterations [34]. Besides, in the initial phase of the denoising process, the primary emphasis is on refining the structure of the restored image, while in the later stages, the main focus shifts to refining the intricate textures of the restored image [35]. To improve image restoration quality, it is not sufficient to perform feature extraction on the guide image only once, nor is it adequate to rely solely on simple addition or cross-attention to fuse the features.

Different from aforementioned methods, to ensure the fidelity, authenticity, and identity consistency of restored images in complex degradation scenarios, we propose a 3D prior-guided diffusion model by incorporating 3D facial prior information as constraints. Besides, we propose a multi-level feature extraction module to extract structural and identity information from 3D prior information at each time step and then weight-adaptively fuse this conditional guidance information with noisy image feature information through a Time-Aware Fusion Block. Overall, the main contributions of this paper can be summarized as follows:

- A novel diffusion-based face restoration network is proposed, which integrates 3D facial structure into the noise estimation process. In our approach, a multi-level feature extraction method is tailored to extract both structural and

identity information, which is then projected into the latent noise space.

- To well adapt to the denoising process of the diffusion model that begins with structure refinement and progresses to texture detail enhancement, a Time-Aware Fusion Block (TAFB) is proposed to effectively and weight-adaptively fuse facial prior features and noisy image features at different time steps.
- Comprehensive experiments demonstrate that the proposed method performs favorably against state-of-the-art algorithms on synthetic and real-world datasets in terms of image quality restoration and identity consistency.

## 2 RELATED WORK

### 2.1 Blind face restoration

As a fundamental task in computer vision, blind face restoration (BFR) aims to recover a high-quality face image from its degraded counterpart in the presence of unknown degraded types and parameters. CNN-based approaches can learn mapping relationships between low-/high-quality image pairs from large-scale collected datasets, but they often fail to restore high-frequency texture details on the face. In recent years, significant progress has been made for BFR by using facial geometry priors and generative priors. Harnessing the specific structure and details of faces, techniques grounded in facial geometry priors utilize prior information such as parsing maps, landmarks, and reference images to enhance the restoration of facial images [3, 4, 7]. On the other hand, generative priors-based methods employ powerful generative models like StyleGAN [20] and incorporate adversarial training to enhance the visual quality of the restored images. Despite the fact that generative prior methods have the ability to restore facial details more effectively when compared to CNN-based restoration approaches, they may encounter significant challenges pertaining to training difficulties and model collapse problems [45].

### 2.2 Diffusion model

In the past few years, the majority of generative tasks in the field of computer vision have been dominated by GAN-based methods [17, 33], which generate decent images through adversarial training. However, these methods may encounter challenges in training difficulties and model collapse [45]. With the application of diffusion models [14, 28] in the generative task domain, these models have demonstrated unprecedented generative capabilities in terms of image quality and diversity. Diffusion models have also been widely applied to various computer vision tasks, including image super-resolution [21, 30, 32], image inpainting [27], image segmentation [1, 2], image-to-image translation [31], text-to-image translation [10], and more. In the context of BFR, DiffBIR [25] Utilizes the pre-trained text-to-image diffusion model and adopts a multi-stage pipeline approach for image restoration. However, it is characterized by a substantial number of parameters and relies solely on addition for feature fusion. To address the challenge of incorporating diverse degradations for real-world scenarios in training data, the Diffusion-based Robust Degradation Remover (DR2)[40] introduces a method that transforms degraded images

into degradation-agnostic predictions before utilizing an enhancement module for high-fidelity image restoration.

Denoising diffusion probabilistic models (DDPM) consist of forward and backward Markov processes. The forward process gradually adds random noise to the image, and we denote these latent variables as $x_1, ..., x_T$, where $x_T$ becomes a completely noisy image. The backward process of DDPM is a denoising process, where it learns a Markov chain that gradually transforms a simple noise distribution (such as isotropic Gaussian distribution) into the target data distribution. Throughout the entire forward and backward processes of DDPM, the dimensions of the image remain consistent. Each step of the forward process can be represented by the following equation:

$$q\left(x_t \mid x_{t-1}\right) := N\left(x_t; \sqrt{1-\beta_t}x_{t-1}, \beta_t \mathbf{I}\right), \qquad (1)$$

where $\beta_1, ..., \beta_T$ are fixed variance values. At each step, Gaussian noise with variance $\beta_t$ is added, resulting in the final $x_T$ being mapped to pure Gaussian noise. Let $x_0$ be the original image, and it is possible to obtain the noisy image at any step $t$ based on $x_0$:

$$q\left(x_t \mid x_0\right) := N\left(x_t; \sqrt{\bar{\alpha}_t}x_0, (1-\bar{\alpha}_t)\mathbf{I}\right), \qquad (2)$$

where $\alpha_t := 1 - \beta_t$ and $\bar{\alpha}_t := \prod_{n=1}^{t} \alpha_n$. These parameters are predefined prior to training. DDPM achieves the denoising process by predicting the mean of the noise added from step $x_t$ to $x_{t-1}$:

$$p_\theta\left(x_{t-1} \mid x_t\right) = N\left(x_{t-1}; \mu_\theta\left(x_t, t\right), \sigma_t^2 \mathbf{I}\right), \qquad (3)$$

where $p_\theta\left(x_{t-1} \mid x_t\right)$ represents the backward process from $x_t$ to $x_{t-1}$, while $\mu_\theta\left(x_t, t\right)$ denotes the diffusion model with parameter $\theta$. At step $t$, $x_{t-1}$ can be expressed by the predicted mean and $x_t$:

$$x_{t-1} = \frac{1}{\sqrt{\alpha_t}}\left(x_t - \frac{1-\alpha_t}{\sqrt{1-\bar{\alpha}_t}}\mu_\theta\left(x_t, t\right)\right) + \sigma_t \mathbf{z}, \qquad (4)$$

where $\mathbf{z} \sim N(0, \mathbf{I})$ is a standard Gaussian noise and has the same dimensionality as noisy image $x_1, ..., x_T$. By performing denoising for T steps, the pure noisy image can be transformed into the target data distribution.

## 3 PROPOSED METHOD

The overall framework for face restoration by incorporating 3D priors into a diffusion model is illustrated in Fig. 4. Our overall framework mainly consists of two branches, including the 3D reconstruction branch and the diffusion branch. The 3D reconstruction branch includes the SwinIR model and the 3DMM model, while the denoising diffusion branch mainly includes a U-net, a multi-level feature extraction module, and a Time-Aware Fusion Block (TAFB). The low-quality image is initially restored using the pre-trained restoration module, SwinIR, resulting in an initial face restoration result denoted as $\mathbf{x}_{init}$. The 3D facial image is reconstructed using the D3DFR method [6]. The multi-level feature extraction module extracts identity and structural information features across different scales of the 3D facial image. These features are then input into the TAFB, in conjunction with features extracted from the noisy image and the current timestep $t$. Subsequently, the time-aware block fuses the features across various time steps and passes them

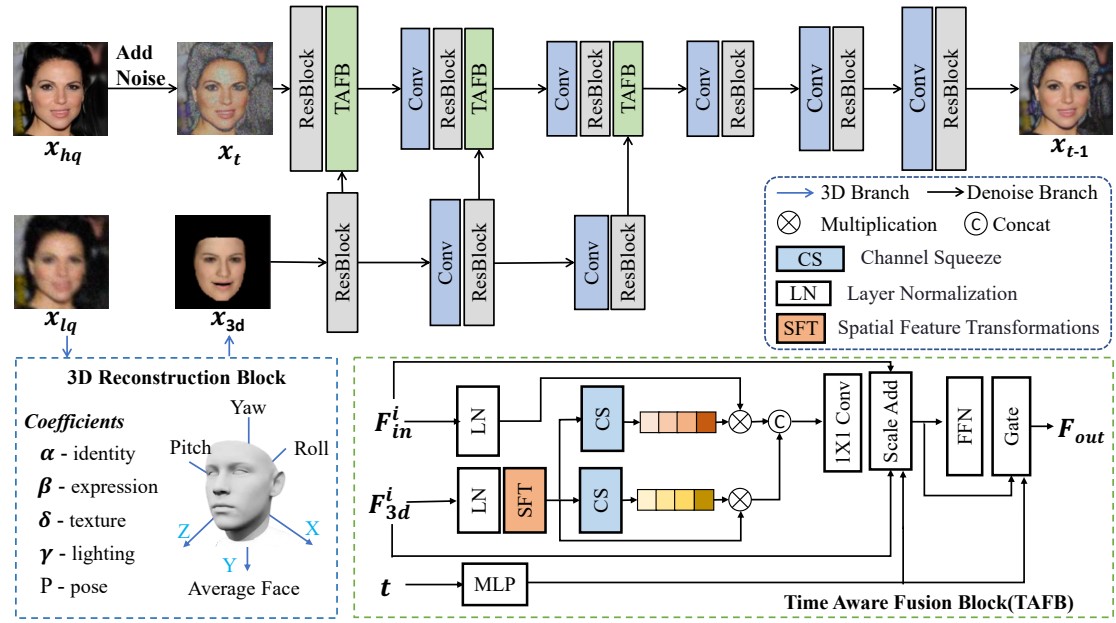

**Figure 4: An overview architecture of 3D priors embedded diffusion model. Top: Our framework consists of two parts: the 3D reconstruction block and the denoising diffusion branch. Bottom: The TAFB module fuses 3D features with features extracted by the denoising network.**

to the subsequent feature extraction block. We will introduce the 3D reconstruction branch and the diffusion branch in detail.

## 3.1 Motivation and novelty

Although current diffusion-based blind face restoration methods have shown promising results in terms of image quality restoration, they often fail to ensure the identity consistency of the restored faces, as illustrated in Fig. 2. It is mainly due to the conflict and balance of realism and fidelity, particularly in complex degradation scenarios. To mitigate this problem, 3D facial priors as structure and identity constraints are embedded into a denoising diffusion process to keep the high fidelity while generating high-realism images. However, considering the dynamic nature of the denoising process in the diffusion model, which involves initial structure refinement followed by texture detail enhancement, we design ingenious modules to incorporate the 3D priors into the diffusion model. *1.)* A customized multi-level feature extraction method is designed to fully exploit both structural and identity information of 3D facial images, which are then mapped into the noise estimation process. *2.)* A Time-Aware Fusion Block (TAFB) is proposed to enhance the fusion of identity information into the noise estimation and offer a more efficient and adaptive fusion of weights for the denoising process. *3.)* The features of 3D facial priors are multi-iteratively embedded into the denoising process at each step and avoid inadequate excavation of 3D facial priors.

## 3.2 3D reconstruction block

3D face priors encompass a wealth of hierarchical features, including low-level details such as sharp edges and lighting, as well as perceptual-level information related to identity. However, Low-quality images for blind face restoration often suffer from multiple complex types of degradation (e.g., blur, noise, JPEG compression artifacts, low resolution, etc.). So the input low-quality image $x_{Lq}$ is first processed by the pre-trained restoration module, SwinIR [24], to obtain the initial face restoration result $x_{Init}$.

$$x_{init} = \mathcal{F}_{\text{SwinIR}}\left(x_{Lq}\right), \tag{5}$$

The outcomes obtained will be utilized as inputs for the diffusion model and the 3D Morphable Model (3DMM) [6] module. The low-quality image is processed using ResNet-50 [12] to obtain the 3DMM coefficients $z_{3d}$.

$$z_{3d} = \mathcal{G}_{\text{Res50}}\left(x_{init}\right), \tag{6}$$

where $z_{3d}$ is a 257-dimensional vector, denoted as $z_{3d} = (\boldsymbol{\alpha}, \boldsymbol{\beta}, \boldsymbol{\delta}, \boldsymbol{\gamma}, \mathbf{p})$, where $\boldsymbol{\alpha}, \boldsymbol{\beta}, \boldsymbol{\delta}, \boldsymbol{\gamma}$ and $\mathbf{p}$ respectively represent the identity information, facial expression, texture, illumination [29], and facial pose. We define the 3D shape $\mathbf{S}$ and texture $\mathbf{T}$ of a face as follows:

$$\mathbf{S}(\boldsymbol{\alpha}, \boldsymbol{\beta}) = \overline{\mathbf{S}} + \mathbf{B}_{id}\boldsymbol{\alpha} + \mathbf{B}_{exp}\boldsymbol{\beta} \tag{7}$$

and

$$\mathbf{T}(\boldsymbol{\delta}) = \overline{\mathbf{T}} + \mathbf{B}_t\boldsymbol{\delta}, \tag{8}$$

$\overline{\mathbf{S}}$ and $\overline{\mathbf{T}}$ respectively represent the average facial shape and facial texture. The variables $\mathbf{B}_{id}$, $\mathbf{B}_{exp}$ and $\mathbf{B}_t$ represent the principal component analysis (PCA) bases for identity, expression, and texture, respectively. The color information of 3D face, denoted as $\mathbf{C}$, can be represented as:

$$\mathbf{C}(i) = \mathbf{c}_i\left(\mathbf{n}_i, \mathbf{t}_i, \gamma\right) = \mathbf{t}_i \cdot \sum_{b=1}^{27} \gamma_b \Phi_b\left(\mathbf{n}_i\right), \tag{9}$$

The 3D image is reconstructed using the D3DFR method, with coefficients $\mathbf{S}$, $\mathbf{T}$, and $\mathbf{C}$. The preliminary restoration module and the 3D reconstruction module are not involved in the training process.

As illustrated in Fig. 3, 3D faces reconstructed directly from low-quality images exhibit significant deviations in facial expressions, structure, facial features, mouth shape, and illumination. In contrast, the 3D faces we reconstructed appear more realistic and faithful to the original subject.

## 3.3 Denoising diffusion branch

In the denoising diffusion branch, we initially subject $x_{hq}$ to the forward process to derive the noise image as per Eq. 2, (where $t \in [1, T]$, and $T$ represents the total number of denoising diffusion steps). Subsequently, the reconstructed facial 3D image is fed into the multi-level feature extraction module to extract features ranging from coarse to fine, thereby capturing both structural and identity information within the facial 3D image.

$$F_{3d}^1 = \text{ResBlock}(x_{3d}), \qquad (10)$$

ResBlock uses the same structure as SR3 [32], and the obtained features will be sent to the next level for feature extraction.

$$F_{3d}^i = \text{ResBlock}\left(Conv_{3\times3}\left(F_{3d}^{i-1}\right)\right), \qquad (11)$$

Downsampling is performed through a convolution layer, where $F_{3d}^i$ represents the facial 3D image features extracted from the $i$-th layer. We also use ResBlock for feature extraction on the noisy image $x_t$. $F_{in}^i$, $F_{3d}^i$, and timestep $t$ is input into the Time Aware Fusion Block, as depicted in Fig. 4. Since the guidance information required by the denoising diffusion model is different at different time steps, we first input t into the Multilayer Perceptron (MLP) to learn the weight parameters.

$$\alpha 1, \beta 1, \gamma 1, \gamma 2, \gamma 3 = MLP(t), \qquad (12)$$

We apply Spatial Feature Transform (SFT) to modulate the facial 3D image features processed by the LayerNorm layer.

$$\begin{aligned} F_1 &= SFT\left(\text{LayerNorm}\left(F_{3d}^i\right), \alpha 1, \beta 1\right) \\ &= \alpha 1 \odot \left(1 + \text{LayerNorm}\left(F_{3d}^i\right)\right) + \beta 1, \end{aligned} \qquad (13)$$

where $\odot$ represents Element-wise Multiplication, passing $F_1$ through different Channel Squeezes (CS) can obtain different weights in the channel dimension. For the sake of simplicity, we only draw four channels in Fig. 4, two sets of different weights. Multiply with $F_1$ and $F_{in}^i$ passing through the LayerNorm layer, respectively. Through channel attention, the model can focus more on important structural information in facial images.

$$\begin{aligned} F_3 &= CS_1(F_1) \odot F_1, \\ F_4 &= CS_2(F_1) \odot \text{LayerNorm}\left(F_{in}^i\right), \end{aligned} \qquad (14)$$

After splicing $F_3$ and $F_4$ through concat, we use 1X1 convolution to transform the number of channels, and then pass through the scale add module. This module uses the weights learned at time step $t$ to combine the output features, denoising image features, and 3D Features.

$$F_5 = \text{Conv } 1 \times 1(\text{Concat}(F_3, F_4)) + \gamma_1^* F_{in}^i + \gamma_2^* F_{3d}^i, \qquad (15)$$

Then $F_5$ is input to the Feedforward Neural Network (FFN) and then passes through the Gate layer. The gate layer is mainly used as a gating mechanism in the feedforward network, and the weights are learned by $t$.

$$F_{out} = FFN(F_5) + \gamma_3^* F_5, \qquad (16)$$

We optimize the conditional denoising diffusion model through the following equation:

$$L_{DM} = E_{x,\epsilon \sim \mathcal{N}(0,1),t}\left[\|\epsilon - \mu_\theta(x_t, x_{3d}, t)\|_2^2\right]. \qquad (17)$$

In the inference stage, we use the same truncated sampling method as [43] for inference. We set the denoising steps to 100, and the specific network architecture layers will be presented in the supplementary materials.

## 4 EXPERIMENTS

### 4.1 Experimental settings

**Datasets.** Following the methods [22, 37, 42], we also selected the FFHQ [19] dataset as our training dataset, which consists of 70,000 high-resolution face images with a resolution of 1024×1024. Initially, we used simple downsampling to resize the images in the dataset from $1024 \times 1024$ to $512 \times 512$, which served as the high-quality (HQ) images in our training dataset. Following the methods [22, 37, 42], we also employed the same random degradation approach to synthesize the LQ images:

$$x = [(y \otimes k_\sigma) \downarrow_r + n_\delta]_{\text{JPEG}_q}, \qquad (18)$$

The corresponding LQ images were synthesized via Eq. (18), where $y$ represents the HQ image, $k_\sigma$ denotes the Gaussian blur kernel, $r$ signifies the downsampling factor, and $q$ represents JPEG compressed images with a quality factor of $q$. We randomly sampled the parameters $\sigma, r, \delta, q$ from $\{0.1 : 10\}$, $\{4 : 20\}$, $\{1 : 20\}$, $\{30 : 70\}$.

We utilized three datasets for evaluation: CelebA-Test [18], LFW-Test [16], WIDER-Test [41], and WebPhoto [37]. The CelebA-Test dataset consists of 3000 synthetic images randomly sampled from CelebA-HQ [18], a high-resolution image dataset. The LQ images represent degraded images with an unknown degradation model and parameters via Eq. (18). The LFW-Test dataset comprises 1711 face images collected from the internet, representing real-world data with a certain level of complexity and diversity. The WebPhoto-Test dataset consists of 407 face images gathered from various online sources. The WIDER-Test dataset comprises 970 severely degraded facial images sourced from the WIDER Face dataset [41].

**Implementation details.** In the experiment, we first validated the effectiveness of the proposed method on the blind restoration task. Then, we further demonstrated its superiority by testing on synthetic and real datasets. Our method employed the Adam optimizer. The default initial learning rate is set to 0.0001, and the learning rate do not decay during training. The experiment is conducted on the A100 GPUs with a batch size of 16.

**Quantitative comparisons of CelebA-test dataset.** The results obtained from our experiments, as presented in Tab. 1, demonstrate that our approach outperforms other methods in terms of quantitative measures when evaluated on the CelebA-Test dataset. Our approach achieves the highest scores in FID-F, FID-G, and IDS, indicating that our restoration results closely resemble both the

**Figure 5: Qualitative comparisons of blind face restoration methods on the CelebA-Test dataset[18]. Our method performs better in both identity consistency and structure consistency.**

| Metrics | PSFRGAN [3] | GFPGAN [37] | Codeformer [46] | VQFR [11] | DiffBIR [25] | DR2 [40] | Ours |
|---------|-------------|-------------|-----------------|-----------|--------------|----------|------|
| PSNR ↑  | 21.0868     | 21.7811     | 22.0322         | 21.1516   | 22.0539      | 21.2123  | 22.3251 |
| SSIM ↑  | 0.5535      | 0.6236      | 0.5880          | 0.6073    | 0.5963       | 0.6160   | 0.6327 |
| LPIPS ↓ | 0.4021      | 0.4156      | 0.3197          | 0.3196    | 0.3495       | 0.4013   | 0.3301 |
| FID-F ↓ | 57.96       | 95.36       | 58.48           | 53.45     | 47.08        | 75.00    | 45.77 |
| FID-G ↓ | 53.34       | 68.36       | 22.81           | 21.10     | 23.20        | 48.52    | 19.36 |

**Table 1: Quantitative evaluation of blind face restoration on CelebA-Test dataset [18] using 3000 randomly selected images. The best and second-best results are highlighted in red and blue, respectively.**

| Datasets | Metric | LQ | PSFRGAN [3] | GFPGAN [37] | Codeformer [46] | VQFR [11] | DiffBIR [25] | DR2 [40] | Ours |
|----------|--------|----|-------------|-------------|-----------------|-----------|--------------|----------|------|
| LFW-Test [16], | | 128.1278 | 51.1954 | 54.9781 | 54.0855 | 51.1867 | 40.6470 | 53.7616 | 45.6765 |
| WebPhoto [37], | FID↓ | 172.1109 | 88.3238 | 120.7289 | 85.9605 | 88.0733 | 94.2337 | 124.1867 | 83.8059 |
| WIDER-Test [41] | | 201.4464 | 51.5343 | 51.7469 | 39.9693 | 38.7984 | 33.1132 | 54.1510 | 36.7571 |

**Table 2: For the performance of blind face restoration, we conducted tests on the real datasets LFW-Test [16], WebPhoto [37], and WIDER-Test [41] using the Fréchet Inception Distance (FID) as the evaluation metric. The best and second-best results are highlighted in red and blue, respectively.**

distribution of real face images and natural images, while also maintaining the structural consistency and identity consistency of the restored face. Besides, the pixel-level metrics SSIM and PSNR are metrics used to evaluate structural similarity and image quality. In these two indicators, the performance of our method also reaches the top scores among all methods, indicating that the restored face image is the best in terms of structural similarity and image quality.

**Quantitative comparisons of the real-world dataset.** Furthermore, our study encompasses a comprehensive quantitative analysis of the real-world datasets LFW-Test, WebPhoto, and WIDER-Test as delineated in Tab. 2. Notably, our method emerged as the top performer on the WebPhoto dataset, showcasing its exceptional efficacy in facial restoration. Conversely, our method secured the second-highest performance on the LFW-Test dataset, as evidenced by the FID evaluation metric. However, real-world images typically

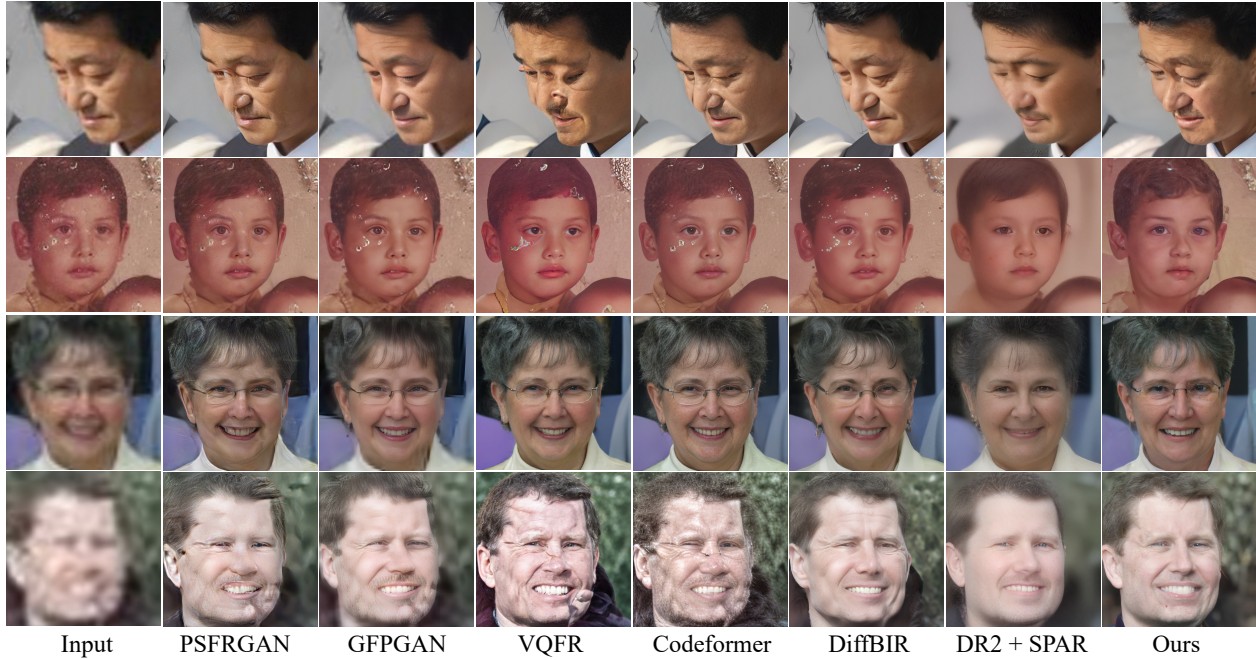

| Input | PSFRGAN | GFPGAN | VQFR | Codeformer | DiffBIR | DR2 + SPAR | Ours |

Figure 6: Qualitative comparisons of blind face restoration methods on the real-world datasets. The results in the first row are from the LFW-Test dataset [16], the results in the second row are from the WebPhoto dataset [37], and the results in the third and fourth rows are from the WIDER-Test dataset [41].

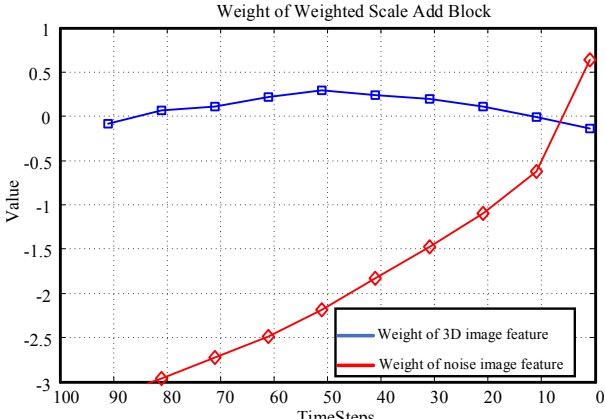

Figure 7: Analysis of the weight in Scale Add Block.

exhibit lesser degradation compared to synthetically altered images. Consequently, the full extent of our method's optimal facial repair capabilities may not be fully realized in such scenarios.

**Qualitative comparisons of CelebA-test dataset.** We conduct a qualitative analysis of six blind face restoration methods. As shown in Fig. 5, the highlighted regions in red boxes indicate significant differences among the methods in terms of facial detail restoration. GFPGAN [37], a GAN-based approach, fails to recover in severe degradation cases. On the other hand, CodeFormer [46] learns discrete codebooks and decoders to store high-quality visual parts of face images through self-reconstructive learning. Our method, by incorporating 3D facial prior information into the diffusion model, better ensures the preservation of facial identity. Our approach

demonstrates excellent fidelity in facial contours, nose, eyes, and mouth, approaching the ground truth.

**Qualitative comparisons of Real-World dataset.** The qualitative outcomes of the real-world dataset are depicted in Fig. 6. Existing methodologies struggle to compensate adequately for information when the input image experiences extensive degradation. In contrast, our methodology introduces 3D facial prior information, resulting in visually more appealing outputs, particularly in cases of severe degradation in the input image.

**Analysis of the weight in Scale Add Block.** As shown in Fig. 7, the weight of 3D image features in Scale Add Block begins to increase as denoising proceeds. The first stage is mainly the recovery of structural information, and the subsequent stages are the recovery of detailed texture information [35]. The process will destroy certain structural information, and the weight of 3D prior information will be reduced. The second stage mainly relies on denoising image features, and the weight continues to increase.

**Analysis of computational complexity and inference time.** The comparative computational complexity analysis between our method and existing diffusion-based techniques is outlined in Tab. 4. Parameter count and Multiply-Accumulate Operations (MACs) are calculated using the profile function sourced. Inference time measurements are performed using a single NVIDIA A100 GPU.

## 4.2 Ablation study

**Metrics.** We evaluate our method using three distinct perceptual metrics: LPIPS [44], and FID (Fréchet Inception Distance) [13]. For completeness, we also include two distortion-based metrics: PSNR

| LQ | (a) | (b) | (c) | (d) | (e) | GT |

**Figure 8: Ablation study on the CelebA-Test dataset [18]. (a) without a facial 3D image as the guidance condition, (b) without TAFB as a fusion block, (c) using TAFB without time Embedding, (d) replacing 3D facial reconstruction images with initial recovery results, and (e) our model. Our approach exhibits superior performance in recovering facial details.**

| Ablation Strategy | FID-G↓ | LPIPS↓ | PSNR↑ | SSIM↑ |
|---|---|---|---|---|
| (a) w/o facial 3D Prior | 28.25 | 0.3536 | 22.5519 | 0.60 |
| (b) w/o TAFB | 24.40 | 0.3744 | 22.5482 | 0.59 |
| (c) w/o Time Embedding | 25.13 | 0.3585 | 22.5077 | 0.60 |
| (d) with initial recovery results | 21.50 | 0.3404 | 22.20 | 0.62 |
| (e) Ours | 19.21 | 0.3321 | 22.6997 | 0.62 |

**Table 3: Ablation study on the CelebA-Test dataset. The red and blue indicate the top-performing and second-best results, respectively.**

| Method | Parameters | MACs | Time(s/image) |
|---|---|---|---|
| DR2 [40] | 179.31M | 918.843G | 0.694 |
| DiffBIR [25] | 1244.63M | 1070.71G | 6.786 |
| Ours | 180.51M | 308.415G | 6.687 |

**Table 4: The computational complexity comparison between our method and other diffusion-based methods. The red and blue indicate the top-performing and second-best results, respectively.**

and SSIM [38]. In particular, when testing FID on the CelebA-Test dataset, the FFHQ dataset is used as reference data.

**Ablation settings.** To demonstrate the benefits of incorporating 3D embeddings into the diffusion model for capturing more facial feature information while ensuring identity consistency in the restored images, we conduct ablation experiments on our proposed method, considering five experimental groups: (a) removing facial 3D images as guiding conditions and eliminating the multi-level feature extraction module and the TAFB module; (b) excluding TAFB as a fusion block and using concatenation to combine facial 3D images and noisy images, only altering the input channel numbers of the diffusion model; (c) employing a TAFB without time embedding; (d) removing the 3DMM reconstruction module and directly inputting the output of SWINIR into the multi-level feature extraction module; and (e) our proposed model.

**Quantitative and qualitative analyses of ablation settings.** We conduct both quantitative and qualitative analyses on the CelebA-Test dataset [18]. As depicted in Fig. 8, Our method shows better recovery effects on individual glasses, eyes, mouth, eyebrows, and facial structures. Since the 3D prior information can reconstruct the area under the glasses, when the 3D prior information is missing, the eyes will fail to reconstruct. If the TAFB module fusion function or time embedding is not integrated, the generated image is prone to over-smoothing because conditional guidance cannot provide different guidance information at different time steps.

Based on the findings from the quantitative analysis presented in Tab. 3, it is evident that the absence of facial 3D prior information as a guide results in the most significant decline across all indicators. This is attributed to the crucial role of facial 3D prior information

in offering a clear facial structure for face restoration and identity preservation. Notably, the omission of the temporal embedding block in the fusion method leads to a substantial decrease in the quality of the restored image. This is due to the inability to provide more precise guidance throughout the entire denoising process. Directly guide the diffusion model through the initial restored image for face restoration. Since there is no 3D facial reconstruction of the eye area, the eye area will encounter serious artifacts.

## 5 CONCLUSION

We propose a blind degraded face image restoration model based on a 3D facial prior diffusion model, which is inspired by the fact that the 3D prior information not only contains facial details but also includes identity information. To ensure realism and fidelity, 3D priors can be regarded as identity and structure constraints in the denoising diffusion process to ensure identity consistency while generating high-quality images. Specifically, the structural features and identity features in the 3D prior information are extracted through the multi-level feature extraction module. Given that the denoising process of the diffusion model primarily involves initial structure refinement followed by texture detail enhancement, we propose a Time-Aware Fusion Block (TFAB). TAFB is used to effectively and weight-adaptively fuse the features with the features extracted from the noise image to more accurately predict the noise and restore the identity and structure consistent with face images. Extensive experiments demonstrate that the proposed model performs favorably against state-of-the-art algorithms.

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

Received 20 February 2007; revised 12 March 2009; accepted 5 June 2009

