# OpenReview forum: "3D Priors-Guided Diffusion for Blind Face Restoration"
_acmmm.org/ACMMM/2024/Conference — MM2024 Poster_

### Official Review · Reviewer_Vhuj · 2024-05-12

**Rating:** 4
**Confidence:** 4

**Summary:**

This paper uses 3DMM and Diffusion Model for Blind Face Restoration. Specifically, this work uses DiffIR to pre-restore noisy images for better 3DMM, and then fuses 3DMM to diffusion model using a SFT-like method.

**Strengths:**

1. Recovering 2D imgs with 3D priors is a difficult task, especially when the 3D info is not just accurate.
2. As far as I know, the first Diffusion-based BFR method ultilizing 3D priors.

**Limitations:**

1. This paper mentions ensuring id consistency. However, I haven't seen any innovative explicit ID enhancement operations, as fusing 2D feature maps with a SFT module is a quite common operation.
2. It seems that other methods have better results when the image is not so noisy. eg. the first 2 rows in Fig. 6.
3. Is the downsampling/blurring factor the same as previous methods during training? If different, will it lead to unfair issues?
4. Will the information provided by the 3D part cause interference for images that are not so blurry?

**Suitability:**

3

---

### Official Review · Reviewer_3ERu · 2024-05-22

**Rating:** 6
**Confidence:** 4

**Summary:**

This paper proposes a novel blind face restoration network for recovering high-resolution face images from degraded face images. The main contributions include innovatively integrating 3D face structure into the noise estimation process and designing a multi-level feature extraction method to extract structural and identity information. Additionally, a temporal-aware fusion block is proposed to adapt the denoising process of the diffusion model from structure refinement to texture detail enhancement. The paper has good innovation and meets the requirements of the conference.

**Strengths:**

This paper clearly articulates the key issues that need to be addressed in blind face restoration and gradually introduces the proposed method, providing a novel perspective for the research field of blind face restoration.

**Limitations:**

(1) There are some issues with the sentences in the paper, for example, in Section 4.1: ‘We utilized three datasets for evaluation: CelebA-Test [18], LFW-Test [16], WIDER-Test [41], and WebPhoto [37].’ Here, the authors actually used four test datasets, but the statement says three.
(2) The proposed method in this paper performs exceptionally well in preserving facial structure. Do the authors have further ideas regarding the details of facial features, such as eyes and makeup?

**Suitability:**

2

---

### Official Review · Reviewer_c2BF · 2024-05-26

**Rating:** 4
**Confidence:** 3

**Summary:**

The paper titled "3D Priors-Guided Diffusion for Blind Face Restoration" introduces a novel framework for restoring clear face images from degraded counterparts. The approach leverages a diffusion-based model that incorporates 3D facial priors as structural and identity constraints into a denoising diffusion process. The authors propose a Time-Aware Fusion Block (TAFB) to adaptively fuse facial prior features with noisy image features at different time steps of the denoising process. The method is evaluated on both synthetic and real-world datasets, demonstrating improved performance over state-of-the-art algorithms in terms of image quality restoration and identity consistency.

**Strengths:**

1.The paper presents a unique approach to blind face restoration by integrating 3D facial structure and identity information into a diffusion model, which is a novel contribution to the field. The proposed method's technical correctness is evident in its detailed explanation of the diffusion process and the innovative Time-Aware Fusion Block, which shows a deep understanding of the problem space.  The paper provides a comprehensive evaluation using both synthetic and real-world datasets, which strengthens the credibility of the proposed method's effectiveness.
2.The paper is well-structured, with clear explanations and figures that effectively illustrate the proposed method and its results.
3. The paper is supported by a solid theoretical foundation and backed by empirical evidence through extensive experiments.

**Limitations:**

1. While the paper discusses the computational complexity, a deeper analysis comparing the proposed method with other state-of-the-art techniques in terms of runtime and resource efficiency could be beneficial.
2. The author presented five different experimental groups in the ablation experiment section to evaluate the proposed method. Near P534, the author stated that during the inference stage, the same truncation sampling method as in reference [43] was used. I don't know if the author is considering increasing the use of reasoning methods other than truncation sampling, perhaps the experimental results will be better.
3. The paper could benefit from additional discussion on the model's robustness to different types and degrees of degradations.
4. P556, should "We utilized three datasets..." be changed to "We utilized four datasets..."?
5. Should the full names of FID-F and FID-G evaluation indicators be provided in section 4.1.
6. The paper primarily focuses on face restoration. It would be useful to see how the proposed method generalizes to other image restoration tasks.

**Suitability:**

3

---

### Meta-Review · Area_Chair_9AEb · 2024-07-04

**Recommendation:** Accept (Poster)
**Confidence:** 5

**Metareview:**

All 3 reviewers suggested acceptance for this paper (1 Weak Accept and 2 Borderline Accept). The AC has read the paper, reviews, and rebuttal, and concurs that this paper should be accepted, given the technical contribution of injecting 3D prior into face restoration diffusion model and the good performance the proposed method. Congratulations and please revise the paper according to the reviewer's suggestions. The author should also consider citing a relevant work:
Wen et al. Face Video Deblurring using 3D Facial Priors. ICCV 2019.